# Design and Application of Two Consistency Verification Methods for Weather Radar Networks in the South China Region

Heng Hu<sup>1</sup>, Shunqiang Pei<sup>2\*</sup>, Lei Wu<sup>1</sup>, Nan Shao<sup>1</sup>, Chunyan Zhang<sup>3</sup>, Hao Wen<sup>1</sup>, YunLei Liu<sup>1</sup>

- <sup>1</sup>Meteorological Observation Center of China Meteorological Administration, Beijing 100081, China;
- <sup>2</sup> Public Meteorological Service Center, China Meteorological Administration, Beijing 100081, China;
  - <sup>3</sup> Guangdong Meteorological Data Center, Guangzhou 510640, China;

Correspondence to: Shunqiang Pei (peishunqiangcma@163.com)

Abstract. The observational consistency between ground-based weather radars significantly impacts the quality of mosaic products and severe convection identification products. The real-time monitoring of observational biases between radars can provide a basis for calibration and validation. This study designed a consistency verification method for weather radar networks based on the FY-3G precipitation radar (SGRCM) and a ground-based weather radar network consistency verification method (AWRCM). From January to October 2024, observational experiments were conducted in the South China region involving 19 S-band weather radars and 13 X-band phased-array weather radars. The aim was to analyze the influencing factors of the consistency verification methods and the observational biases of reflectivity factors for radars with different bands and systems. For the S-band weather radars, the difference in the bias between the two methods ranged from -1.5 dB to 1.4 dB, and the difference in the standard deviation ranged from -1.2 dB to 1.2dB. For the X-band phased-array weather radars, the difference in the bias between the two methods ranged from -6.67 dB to 0.84 dB, and the difference in the standard deviation ranged from -0.38 dB to 1.51 dB. The evaluation results of the two methods show good consistency for weather radars with different bands. We selected one radar with a larger bias for recalibration and rectification, and the changes in bias before and after rectification thus provide a good indication of the improvement in network consistency among the radars.

#### 1. Introduction

Currently, there are 252 new-generation weather radars in operational use across mainland China (137 S-band and 115 C-band radars), with over 300 X-band weather radars. Except for certain mountainous and desert regions in the west, the new-generation weather radars cover most of the densely populated areas of the country. In regions with densely deployed radar sites, there are various degrees of overlap between adjacent radars. It has been observed that, over long-term operational use, radar reflectivity errors are influenced by factors such as an inadequate calibration of radar equipment parameters, beam blocking (Dinku et al.,2002; Liu et al.,2020), clutter interference, and electromagnetic interference in radar rainfall measurement (Travis et al., 2016; Tang et al., 2020; Zhang et al., 2003). These errors result in different observational outcomes from various radars for the same meteorological target due to influences such as the direction of the target, atmospheric conditions, attenuation, obstruction, and clutter. Echo intensity has always been an important parameter for identifying severe convective weather, and it directly determines the accuracy of precipitation products estimated based on the Z-R relationship (Ryzhkov et al.,1995; Fabry et al.,1995; Steiner et al.,1995; Bringi et al.,2001). If adjacent radars observe echo intensity values with discrepancies within overlapping areas during the same observation period, it can affect the quality of radar network

mosaics and increase uncertainty in the assimilation of radar data with other data sources. Therefore, it is crucial to perform a scientific, quantitative analysis of echo consistency in overlapping areas observed by adjacent radars in order to identify and correct observation biases. Some studies have proposed algorithms for evaluating the consistency between adjacent radars and provided a quality control method for matching points (Gourley et al., 2003; Smith et al.,2018; Gao et al.,2020). Zhang Zhiqiang et al. (2008) interpolated radar echoes into a three-dimensional grid to analyze the consistency in the positioning and echo intensity of four radars in the North China region. Vukovic et al. (2014) analyzed the impact of beam blockage in overlap regions between adjacent radars. Wu Chong et al. (2014) and Zhang Lin et al. (2018) conducted comparative studies on the echo differences in consistency between phased-array weather radars and new-generation Doppler weather radars. Xiao Yanjiao and Ye Fei et al. (2020a, 2020b) studied the echo intensity consistency along equidistant lines between adjacent radars based on quality-controlled CAPPI data. However, the CAPPI interpolation algorithm itself introduces biases, which can lead to uncertain sources of error in the network consistency analysis results (Lakshmanan et al., 2006).

Using adjacent ground-based weather radars for a network consistency analysis can more easily identify observation biases in areas with dense radar deployment. However, in regions in the west with sparse radar stations, it may not be possible to match adjacent stations, thus necessitating the use of multi-source observational data for calibration, with precipitation satellite data being a commonly used reference standard. Internationally, the reflectivity factor deviations between satellite-borne precipitation radars, such as TRMM/PR (Tropical Rainfall Measuring Mission/precipitation radar) and GPM (Global Precipitation Mission), and ground-based radars are used to correct radar reflectivity values (Wang et al., 2009; Park et al., 2015; Warren et al., 2018; Protat et al., 2021; Zhi et al., 2023). Domestically, He Huizhong et al. (2002), Wang Zhenhui et al. (2015) compared the consistency between reflectivity measured using the TRMM precipitation radar and ground-based radar echo intensity in China.

45

However, the observational biases and stability of precipitation satellite data can also affect comparison results. Simply calculating quantitative biases between satellites and radars is not meaningful (Bolen et al.,2003, Schwaller et al.,2011). Using precipitation satellite data as a reference standard, transferring the systematic bias between ground-based weather radars and precipitation satellites to the results of a network consistency analysis for ground-based radars can help ascertain the observational biases of radars.

This study selects the South China region as the analysis area, where there is a rich variety of precipitation types and a wide distribution of multi-system and multi-band radars. Developing a multi-source integrated weather radar network consistency analysis method in this region will provide a solid basis for the method's promotion across China. We utilize observational data from China's independently developed FY-3G satellite, obtaining S/C/X-band reflectivity factors after quality control and frequency correction. During satellite overpasses, we perform spatiotemporal matching with ground-based radars to match overlapping areas and analyze deviations. The satellite—ground comparison results are then integrated into the ground-based radar network consistency results to finally determine the reflectivity factor observation biases of the weather radars. This approach provides a quantitative, automated method for the calibration and adjustment of ground-based weather radars.

## 2. Materials and Methods

#### 2.1 Data Introduction



The FY-3G satellite, part of the third batch of FY-3 satellites, was successfully launched on April 16, 2023, from the Jiuquan Satellite Launch Center. The primary payload for precipitation measurement on this satellite is the Precipitation Measurement Radar (PMR), which includes both Ku- and Ka-band radars. This marks the first time China has achieved active satellite-based precipitation detection, with the ability to obtain three-dimensional structural information within precipitation systems. Both radars employ a fully matched scanning mode with a scanning angle of ±20 degrees. The spatial resolution at the nadir point is 5 km, and the vertical resolution is 250 m. The design sensitivity is 18 dBZ for the Ku radar and 12 dBZ for the Ka radar (CMA,2023; Wu,2023a).

This study utilizes Level 2 products from the FY-3G precipitation measurement radar, focusing primarily on the radar reflectivity factors for both the ascending and descending tracks of the Ku radar, corrected for frequency (Wu et al.,2023b). These Level 2 products are provided in a latitudinal and longitudinal grid format ranging from the ground up to 20 km, with a data structure of *nscan\*nray\*nbin*. Here, *nscan* represents the variable number of scan lines, *nray* denotes the number of angle units per scan line, and *nbin* refers to the number of vertical range bins. Fig. 1 shows a brief overview of the descending orbit of the FY-3G precipitation satellite PMR Ku radar.

**Figure 1.** Brief overview of descending orbit of FY-3G precipitation satellite PMR Ku radar (cited from the National Satellite Meteorological Center, The satellite operates in a south-to-north direction.).

During the experiments, we used 19 S-band weather radars and 13 X-band phased-array radars. The distribution of the stations is shown in Fig. 2, and the specific hardware parameters are shown in Appendix A. The ground-based weather radars

use standard format base data. At present, the radars are undergoing mode switching trials and will automatically switch observation modes according to real-time weather conditions: VCP (Volume Coverage Pattern)11 (for convective heavy precipitation), VCP21 (for stratiform precipitation), and VCP31 (for clear skies) (NWS, 2025). We selected 10 stations in the national S/C band weather radar network to conduct a consistency analysis before and after mode switching. The evaluation results from 2024 show that mode switching has no significant impact on the method design involved in this study. The SGRCM uses all elevation angle data from the radar, while the AWRCM only uses the lowest 5 elevation angles, primarily to consider calculation efficiency.


**Figure 2.** Distribution map of ground-based weather radar stations (Blue dots represent S-band weather radars, gray dots represent X-band weather radars)

## 2.2 Method Introduction

#### 2.2.1 Satellite and Ground-Based Radar Comparison Method (SGRCM)

First, the latitude and longitude data from the Geo\_Fields module of the FY-3G Level 2 products are read, which represent grid points on the surface and at an 18 km altitude. Both layers consist of *nscan\*nray* (3892\*59) points. Using *nbin* as the step,

the latitude and longitude for each grid point at every altitude level are calculated. The "height" from the PRE (Data Preprocessing Module) module and the reflectivity factor Ze from the FRE (Frequency Correction Module) module's "zFactorFrequencyCorrectionS" are also read, forming arrays of size 3892\*59\*400 (with a vertical sampling rate of 50 m). These data are then combined to obtain the satellite grid geographical information and reflectivity factor array.

The radar base data are read to generate a three-dimensional array of size m\*n\*k (where m represents the elevation angles, n represents the azimuth angles, and k represents the range bins). Coordinate system transformations are performed from polar coordinates to the first and second reference frames and, finally, to the geodetic coordinate system, which provides the latitude, longitude, and altitude for each range bin (Yang, et al.,2023), along with the reflectivity factor array. The steps for satellite—ground consistency comparison are as follows.

## (1) Spatial and Temporal Collocation

Begin by identifying ground-based radars (GB) whose observational coverage significantly overlaps with the FY-3G PMR (SG) scanning region. Overlap criteria require that at least 3,000 (S/C-band) or 400 (X-band) PMR grid points fall within the GB's observation area. For temporal alignment, only data pairs where the observation times differ by less than 180 seconds are retained.

## (2) Resampling







The FY-3G PMR Ku L2 product is a resampling dataset with 400 bins and a vertical resolution of 50 m, which differs from the original vertical resolution of 250 m used in the SG scanning mode. In this study, the data at each scanning track grid of SG are resampled into a four-dimensional (longitude, latitude, height, time) grid data with a vertical resolution of 250 m (80 bins) and a horizontal track resolution of 5 km, as the SG scanning mode shown in Fig.3. That is, each SG grid is 5 km × 5 km × 250 m. Measurements that are too close to or too far away from the GB stations have significant errors. Through multiple experiments, this study selects the time-paired GB reflectivity data with a distance of 50-150 km away from the stations for S/C-band GBs and 9-42 km for X-band GBs. The GB reflectivity data are then transformed into three-dimensional (longitude, latitude, height) data.

#### (3) Extraction of Stratiform Rain Cases

Stratiform precipitation is isolated using the precipitation classification provided by the SG product at each grid point. Both satellite and ground-based reflectivity values are further restricted to 20-35 dBZ within the 2-4 km altitude range to focus on relatively stable echoes.

#### (4) Pairwise Data Construction

For each spatial-temporal matchup, if multiple GB range bins correspond to a single SG grid cell, they are averaged to produce a composite GB reflectivity value. These paired values SG and averaged GB reflectivity form the basis for subsequent comparison.

#### (5) Consistency Assessment

When at least 20 such matched pairs are available, key statistical indices-namely bias, standard deviation, and correlation coefficient are computed to quantitatively evaluate the consistency between the SG and the GB network.

**Figure 3.** A comparison of the overlap region for the reflectivity factors observed by the FY-3G precipitation satellite and the ground-based S-band weather radar.

Figure 4. Satellite and ground-based radar comparison method diagram.

#### 2.2.2 Adjacent Weather Radar Comparison Method (AWRCM)





The ground-based radar consistency algorithm selects base data from scans with inter-radar distances below a specified threshold (e.g., 300 km for S-band, 100 km for X-band) and volume scan intervals within 3 or 6 minutes, using elevation angles lower than 4.5 degrees from adjacent radars as the data source, which is for considerations of computational efficiency. Terrain data are used to remove occlusions, and non-precipitation echoes are filtered out. For spatial consistency matching, the horizontal and vertical distance thresholds are set to half the shorter path length among the radars; for temporal consistency, the difference in radial observation times must be below a defined threshold (e.g., 60 seconds). Observations with a signal-to-noise ratio less than 15 dB or insufficient horizontal filling at echo boundaries are excluded. The reflectivity threshold is set to 15–35 dBZ (with 35 dBZ serving as the S-band boundary between stratiform and convective precipitation echoes (Yu, 2007). Convective targets are excluded using vertically integrated liquid water (VIL>6.5 kg/m²) (Xiao et al.,2009). Finally, according to the 3-sigma rule, outliers in matched targets within overlapping radar regions are removed, and statistical metrics such as standard deviation and mean bias for the evaluation period are calculated to analyze consistency between adjacent radars.

The spatial consistency matching technique constitutes the main challenge. Wu Chong et al. (2014) and Zhang Zhiqiang and Liu Liping (2011) addressed the challenges of matching S-band phased-array weather radar data with new-generation weather radar data, which result from dissimilar spatial resolutions between the radars. They utilized polar-to-latitude–longitude coordinate transformations, reflectivity spatial interpolation, and other methods to design a spatial matching method for radar data with different resolutions and geographic locations. Zhang Lin et al. (2018) developed a method for the operational new-generation Doppler weather radars, where they transformed the polar coordinates of the first radar into latitude–longitude projection coordinates and searched for targets with consistent projections within the polar coordinates of the second radar. They set altitude thresholds to achieve the spatial matching of data from both radars.

The data addressed in this study pertain to the base data of the new-generation Doppler weather radars in operation. In the spatial matching algorithm, the above-mentioned methods are also employed. The process is described below.

Figure 5. Schematic of two radars' spatial consistency algorithm.

As shown in Fig. 5, let the station coordinates of Radars 1 and 2 be  $(\lambda_I, \phi_1, h_1)$  and  $(\lambda_2, \phi_2, h_2)$ , respectively. For each volume scan data point from Radar 1, the polar coordinates—azimuth  $a_1$ , elevation e1, and slant range  $L_1$  (the red points in Fig. 5 are transformed into latitude, longitude, and altitude  $(\lambda, \phi, H_I)$  using the formulas for converting radar polar coordinates to geographic coordinates and radar altitude calculations. The ground projection point's longitude and latitude are  $\lambda$  and  $\phi$ . The formulas for these calculations are as follows, where  $Km = \frac{4}{2}$ , represents the effective Earth radius factor.

$$\varphi = \sin^{-1}(\cos\beta_1\sin\varphi_1 + \sin\beta_1\cos\varphi_1\cos\alpha_1) \tag{1}$$

$$\lambda = \sin^{-1}\left(\frac{\sin a_1 \sin \beta_1}{\cos \varphi}\right) + \lambda_1 \tag{2}$$

 $\beta_1$  is the angle between the projection point and the center of the Earth at the location of Radar 1.



$$\beta_1 = K_m \tan^{-1} \left( \frac{L_1 \cos e_1}{Rm + h_1 + L_1 \sin e_1} \right) \tag{3}$$

Then, the formula for converting geographic coordinates to radar polar coordinates is used to calculate the data coordinates of Radar 2 under this projection. Radar 2 has multiple scanning elevation angles, and the scanning elevation angle  $e_2$  for a data point is known. Using the coordinate transformation formula, it is straightforward to calculate the polar coordinates—azimuth  $a_2$ , elevation  $e_2$ , and slant range  $L_2$  (the blue points in Fig. 5, with the number of points determined by the intersecting radar 2 radial layers)—as well as their altitude  $h_2$ , based on the conversion from geographic coordinates to radar polar coordinates.

$$\cos \beta_2 = \sin \varphi \sin \varphi_2 + \cos \varphi \cos \varphi_2 \cos(\lambda_2 - \lambda) \tag{4}$$

Here,  $\beta_2$  is the angle between the projection point and the center of the Earth at the location of Radar 2. By using  $\cos \beta_2$ ,  $\sin \beta_2$  can be obtained; thus,

$$\sin a_2 = \frac{\cos \varphi \sin(\lambda - \lambda_2)}{\sin \beta_2} \tag{5}$$

Using Equation 1,  $\cos a_2$  is obtained, and then the azimuth angle  $a_2$  and slant range  $L_2$  are calculated as follows:

$$a_2 = \operatorname{atan2}(\sin a_2, \cos a_2) \tag{6}$$

If  $a_2 

**Figure 6.** The impacts of distance between adjacent radar stations and elevation difference on the distribution of overlapping points.

## 3.1.2 Terrain blockage






When matching adjacent radars, severe terrain blockage in the direction of the overlap points for one of the radars may weaken the radar echo intensity. This can result in significant echo differences at the overlap points between the two radars, leading to inaccurate consistency evaluation results. This issue is not due to the radar itself (Maddox et al., 2002; Bech et al., 2003).

Regarding terrain blockage, Liu Yunlei et al. (2020) utilized SRTM (Shuttle Radar Topography Mission) v4.1 digital elevation data to perform simulations and analyses of beam blockage for the new generation of operational weather radars in China. They sampled the radar detection range, calculated the latitude and longitude and detection height of target points based on radar station information, compared these to topographic data, and used radar altitude formulas and beam widening information to determine beam cross-section blockage at specific elevation angles. This provided beam blockage ratio data (hereinafter referred to as the obstruction rate) for each radar station.

**Figure 7.** Terrain blockage at 0.5° and 1.5° elevation angles for Shantou S-band weather radar station in Guangdong. Figure 7 illustrates the terrain blockage at 0.5° and 1.5° elevation angles for Shantou station in Guangdong. The blockage imarily distributed in the porthwest direction of the station. When analyzing the observation has between Shantou station

is primarily distributed in the northwest direction of the station. When analyzing the observation bias between Shantou station and a adjacent radar located to its northwest, it is necessary to exclude the obstructed radials when calculating the overlap area, as doing so will reduce errors in the network consistency analysis.

#### 3.1.3 Impact of Observation Targets





When the observation target is convective precipitation, the time threshold for calculating observation biases in the overlap areas between the satellite and ground-based radar needs to be limited to a very small range. However, this constraint may not provide a sufficient sample size for statistical analysis. In this study, the target was limited to stable stratiform precipitation, requiring the further classification of precipitation types. In satellite observation data, precipitation classification is performed using two methods: the vertical profile retrieval method and the horizontal pattern method. These methods classify precipitation into three categories: stratiform, convective, and other. The precipitation types identified by these two methods are then consolidated (Wu, 2023).

In the adjacent ground-based radar comparison verification method, we calculated the liquid water content for each grid point. Based on a statistical analysis, we set a threshold (Biggerstaff et al., 2000; Xiao, et al., 2007) to classify observation targets into convective and stratiform precipitation. Fig. 8 shows a consistency comparison of two S-band weather radars (ID5 and ID8) in Guangdong before and after convective filtering. We adjusted the time threshold from 180s to 60s and set the vertically integrated liquid (VIL) threshold to 6.5 kg/m². After filtering, the number of matching points decreased, the correlation coefficient increased from 0.84 to 0.87, the standard deviation decreased from 4.68 dB to 4.34 dB, and the bias changed from -2.19 dB to -2.21 dB. It can be seen that increasing the radial time threshold and VIL filtering improved the correlation and standard deviation in the overlap regions of adjacent radars, although the bias slightly decreased. The reason for this requires further analysis with more accumulated samples.

**Figure 8.** Comparison of two adjacent S-band radars before and after convective filtering. (left: before filtering, right: after filtering.)

# 3.1.4 Impact of Different Bands





BX5 is a standardized X-band weather radar. As a radar to be calibrated, it experiences co-channel interference when operated simultaneously with surrounding X-band radars, necessitating the creation of a blanking zone and the maintenance of its primary observation direction within the first quadrant. Approximately 2 km away from BX5, an S-band dual-polarization weather radar serves as a reference radar. Both radars can scan simultaneously to observe the same precipitation area.

Figure 9 shows the reflectivity factors observed at a 0.5-degree elevation angle by the two adjacent S/X-band weather radars around 14:35 on May 26, 2024, with the X-band radar data not corrected for attenuation. The white box in the figure identifies the same echo region. In the left panel, the reflectivity factor observed by the X-band radar is 10-15 dBZ weaker than that in the right panel observed by the S-band radar. A probability distribution analysis of the reflectivity factors from the overlapping observation areas of the two radars is conducted, as shown on the left side of Fig. 10. The calculated bias, standard deviation, and correlation coefficient are -6.74 dB, 10.12 dB, and 0.18, respectively. We applied an adaptive attenuation correction method (Testud et al.,2000a) to the BX5 radar, and then analyzed the bias between the corrected data and that from the adjacent S-band weather radar. The bias was reduced to -1.68 dB. The X-band radar shows significant attenuation in strong echo areas (Testud et al.,2000b; Bringi et al.,2001). Therefore, in subsequent analyses of the network consistency between X-band and other band weather radars, the reflectivity factor range is set (e.g., 15-35 dBZ), with certain limitations also applied to the signal-to-noise ratio.

**Figure 9.** Comparison of reflectivity factors observed by adjacent X-band (left) and S-band (right) weather radars during a precipitation event on May 26, 2024, at 14:35.

**Figure 10.** Analysis results of reflectivity bias in the overlapping observation area of adjacent X- and S-band weather radars on May 26, 2024, at 14:35 (the horizontal axis (Radar1) represents the X-band radar, while the vertical axis (Radar2) represents the S-band radar). (Left: the results before attenuation correction for the X-band radar. Right: the results after attenuation correction.)

## 3.1.5 Impact of Non-Meteorological Echoes



In radar consistency evaluation algorithms, the impact of non-meteorological echoes at overlapping points must be considered. These echoes may be caused by noise or insufficient target filling, among other reasons. Coastal stations are often

affected by changes in atmospheric refractivity over the ocean (Melsheimer et al., 1998; Skolnik et al., 2008), leading to clear-air echoes or sea clutter, which can significantly influence the comparison results of overlapping areas between adjacent radars.





As illustrated in Fig.11, the two weather radar stations are coastal stations in South China, with an observation time difference of about 2 minutes and a distance of approximately 140 km between them. In the left panel, the third and fourth quadrants exhibit clear-air echoes, while in the right panel, these quadrants display sea clutter echoes. When performing overlap area matching, consistency calculations were conducted for these non-precipitation echoes, resulting in a bias of 8.84 dB. This bias clearly does not stem from radar hardware performance. Therefore, when analyzing the comparison results of overlapping areas between adjacent weather radar stations, it is crucial to first exclude non-precipitation echoes in order to minimize their impact on the statistical outcomes.

**Figure 11.** Impact of clear-air echoes and sea clutter on overlapping area comparison analysis at coastal weather radar stations.

To reduce the impact of noise on the evaluation results, we improved the data filtering method by setting a signal-to-noise ratio threshold (*SNR\_thre* 

Figure 12. Schematic of the degree of beam incomplete filling (The right side of the figure shows the Ref SD results (unit: dB), with a ring set every 100 km and the outermost ring at 460 km.).

Electromagnetic interference can affect the quality of weather radar observation data and the reflectivity factor comparison results between radars (Saltikoff et al.,2016; Nguyen et al.,2017). Fig. 13 (left) shows radial interference occurring at an elevation angle of 0.5 degrees between radial angles of 45 and 52 degrees on the Shantou weather radar (ID5) in Guangdong at 01:30 (UTC) on June 16, 2024. By using a fuzzy logic method (Wen et al.,2020) to eliminate the radial interference, a quality-controlled reflectivity factor map was obtained, as shown in Fig. 13 (right). We extract four physical parameters that characterize radial interference echoes: *DB*, representing the consistency of echo power between adjacent range gates along the radial. *RREF*, representing the spatial extent of the reflectivity factor along the radial. *TDBZ* (units: dB²), representing the texture consistency of the local reflectivity factor along the radial. *SPIN*, representing the sign change of adjacent reflectivity factors within a local region. Based on the probability distributions of these parameters, we construct corresponding membership functions and a binary (0-1) decision criterion for radial interference echoes. The criterion values are then combined via a weighted summation, and any point whose aggregated value exceeds a threshold is identified as a radial interference echo and removed. A consistency analysis comparing the reflectivity before and after interference removal with that of a adjacent S-band weather radar showed that the correlation coefficient, bias, and standard deviation of the two radars improved from 0.88, -1.70 dB, and 4.97 dB to 0.89, -1.68 dB, and 4.92 dB, respectively. This indicates that radial interference reduces the observation consistency between adjacent radars.

**Figure 13.** Reflectivity factor at a 0.5-degree elevation angle for the Guangdong Shantou radar before and after electromagnetic interference quality control: (left) before quality control; (right) after quality control.

**Figure 14.** Consistency analysis results of the Guangdong Shantou radar with a adjacent S-band weather radar before and after electromagnetic interference quality control. (Radar1 is the ID5, and Radar2 is ID8)

#### 3.2 Regional Experimental Results



#### 3.2.1 Evaluation Results of the S-band

In the AWRCM analysis, we set a distance threshold between adjacent radars (for example, 200 km for S-band radars). Any two radars within this threshold can be paired for comparison. Taking Radar 1 as an example, if it can be paired with n surrounding radars, then each volumetric scan will yield n sets of comparison results, and the average of these n results is taken as the final consistency bias value for Radar 1 at that time. If a particular radar has a large systematic bias, this will be reflected

in the bias average. A large standard deviation indicates that the radar's observation results are more dispersed, suggesting a need for further calibration. From January to October 2024, 19 S-band new-generation weather radars in South China were selected to conduct both AWRCM and SGRCM analyses. The differences between the two methods were evaluated using bias and standard deviation as metrics. Fig.15 presents the bias comparison results from both methods. The bias trends are generally similar, with the ground-based consistency analysis showing bias values ranging from -2.06 to 1.65 dB, and a mean of -0.12 dB. The satellite-to-ground consistency analysis produces bias values ranging from -1.28 to 1.13 dB, with a mean of -0.01 dB; notably, the absolute bias is smaller for the satellite-to-ground method than for the ground-based method.




Fig.16 shows the standard deviation comparison for the two methods, mainly concentrated below 4 dB. The differences between the two standard deviations are within  $\pm 1.2$  dB, indicating that both methods provide relatively close assessments of the dispersion of ground-based radar observation bias.

Figure 15. Comparison of mean bias between AWRCM and SGRCM for S-band weather radars in the South China region.

**Figure 16.** Comparison of standard deviations between AWRCM and SGRCM for S-band weather radars in the South China region.




When generating the bar charts for the above statistics, we only selected results where the sample size in the overlapping areas of adjacent radars exceeded 200 to ensure the stability of the results. In the subsequent analysis of single-station, single-time cases, we did not impose this constraint. Data from four selected stations were analyzed. Fig. 17 shows the bias analysis results for ground-based consistency. The gray dots represent the bias of single complete volume scan (5-6min), while the red dashed line represents the mean of bias. Given variations in the number of matched adjacent stations and the weather processes involved, the algorithm computes on a per-time-step volume-scan basis without manual selection of specific weather events; therefore, the analysis sample size varies. The mean of bias between stations ID5, ID8, and their adjacent stations is greater than 0, indicating that these two radars are relatively stronger in the ground-based network, with station ID5 showing a particularly noticeable positive bias. In contrast, stations ID3 and ID2 exhibit negative biases, indicating that these two stations are weaker than their adjacent ground-based radars.

Figure 18 shows the satellite–ground comparison results for the four stations. It can be observed that the reflectivity factor of the FY-3G PMR is generally larger. Among the four stations, station ID5 has the smallest bias, indicating that ID5's intensity trend is consistent with the ground-based analysis.

Figure 17. Bias analysis results of AWRCM for individual S-band weather radar stations.

Figure 18. SGRCM analysis results for individual S-band weather radar stations.

## 3.2.2 Evaluation Results of X-Band Phased-Array Radars



AWRCM and SGRCM analyses were conducted for 13 X-band phased-array weather radars in Guangdong Province, for which attenuation correction has already been applied to the base data (Xiao et al.,2021). The bar chart in Fig. 19 represents the distribution of the mean bias for the two methods. For most phased-array radars, the average bias of both the AWRCM and SGRCM is less than 0, indicating that the reflectivity of the phased-array radars is relatively weaker. This suggests that the attenuation correction applied prior to radar base data generation did not achieve the expected effect, and the reflectivity of X-band phased-array radars remains noticeably weaker compared to S-band radars. The dashed line represents the difference between the AWRCM bias and the SGRCM bias. The results are mainly distributed below 0, suggesting that the bias results from the AWRCM analysis are relatively larger.

**Figure 19.** Comparison of mean bias between AWRCM and SGRCM for X-band phased-array weather radars in the South China region.




The bar chart in Fig. 20 represents the distribution of the standard deviations of the two analysis methods. The results of the AWRCM analysis range from 3.15 to 3.95 dB, while those of the SGRCM analysis range from 1.96 to 4.01 dB, with larger differences in standard deviation observed between different radars in the SGRCM analysis. The dashed line represents the difference between the two analysis results, with most results distributed above 0 dB, indicating that the AWRCM analysis results are relatively larger.

**Figure 20.** Comparison of standard deviations between AWRCM and SGRCM for X-band phased-array weather radars in the South China region.

Two X-band phased-array weather radars, ID22 and ID26, were selected for analysis. Fig. 21 shows the bias analysis results for the AWRCM. The gray dots represent the bias of single complete volume scan (5-6min), while the red dashed line represents the mean of bias. It can be observed that the differences in overlapping observation points between the phased-array

weather radars and the surrounding S-band weather radars are mainly distributed below 0. The mean biases are -2.02 dB and -3.83 dB, respectively, indicating that the X-band phased-array radars are weaker than the S-band solid-state weather radars.




Figure 21. AWRCM analysis results for individual X-band phased-array weather radars.

Figure 22 shows the SGRCM analysis results for the two radars. The bias in the figure represents the reflectivity factor of the phased-array radar minus that of FY-3G, with values of 1.66 dB and 2.11 dB, respectively. This also indicates that the reflectivity factor observed by the X-band phased-array radars is weaker, but the bias results are smaller than the AWRCM analysis results. The SGRCM standard deviation of radar ID22 is smaller than that of radar ID26. From the bias distribution of overlapping observation points, it can be seen that the SGRCM bias of radar ID26 exhibits greater dispersion. A preliminary analysis indicates that the valid SGRCM comparison results for radar ID22 are mainly concentrated in August 2024, whereas those for radar ID26 span June–September. Owing to the longer time window, the precipitation types encountered are more diverse, which may lead to differences in the SGRCM scatter distributions. This conclusion, however, requires further verification and analysis with additional observational data.

From the analysis of radars ID22 and ID26, we observe that although both X-band phased-array radars applied attenuation correction prior to base data generation, the correction performance is not satisfactory. Notable biases remain in the reflectivity factor relative to the adjacent S-band radar. This will increase the complexity of subsequent networked applications of the data; therefore, an additional attenuation-correction step will be introduced before the mosaicking.

**Figure 22.** SGRCM analysis results for individual X-band phased-array weather radars.

#### 4. Discussion






In daily operations, the two consistency evaluation methods provide a basis for real-time monitoring of observational biases in ground-based radars. Once a significant change is detected in the consistency evaluation results, we will initiate the subsequent calibration procedures, including Solar Calibration (Holleman et al., 2022) and Metal Sphere Calibration (Ao, et al., 2022). To determine whether the results of the above consistency analyses are correct and whether they can provide a basis for calibration, we conducted a rectification experiment using an SC (a model of S-band radar) weather radar located in Sanya City, Hainan Province. Before calibration, the deviation between this radar and surrounding radars and that between satellite and ground measurements both exceeded 2.7 dB. During the system calibration, the system parameters of each radar station were revised and recalibrated. The main adjustment involved modifying the transmission branch feeder loss parameter, changing the single-H transmission feeder loss from 1.59 dB to 2.50 dB. By calibrating the internal continuous-wave power using the external continuous-wave power, the internal continuous-wave power before the low-noise amplifier was adjusted from 0.30 dBm before rectification to 1.30 dBm.

Figures 23–24 present the ground-based consistency results and the satellite–ground consistency results before and after calibration. In Fig. 23, the gray dots represent a complete volume scan, and the red dashed line marks August 24, 2024, the date on which the radar site underwent calibration. It can be seen that the deviation between the Sanya radar and surrounding radars exhibits changes before and after calibration. In Fig.24, the panel on the left analyzes the satellite–ground consistency results for the Sanya site from May 4, 2024 to August 20, 2024, while the panel on the right analyzes the satellite–ground comparison results from August 30, 2024 to July 22, 2025. The satellite–ground comparisons indicate that the bias changed

before and after the calibration. However, the stability of the rectification effect still require further verification through the accumulation of long time-series data.

**Figure 23.** AWRCM results of the SC method radar in Sanya City, Hainan Province, before and after rectification. The gray line in the chart represents the variation in deviation, while the red dashed line indicates the rectification time.


**Figure 24.** SGRCM results of the SC method radar in Sanya City, Hainan Province, before and after rectification: left: before calibration; right: after calibration.

The FY-3G PMR Level 2 products have been available since January 2024. Due to the observational characteristics of polar-orbiting satellites, the orbital data over the South China region are limited. Conversely, the X-band phased-array weather radar provides high-frequency observations; however, due to the limited transmission bandwidth, the raw data frequency of the X-band phased-array radar is compressed from 1-minute intervals to 10-minute intervals, resulting in a smaller sample size for analysis. By analyzing the comparison results between the national S-band weather radars and the FY-3G PMR, it was found that the satellite's reflectivity factor is generally stronger, with a mean bias of 0.44 dB. This bias was not considered in the comparison of the results from the two methods. If the satellite–ground consistency results are to be transferred to ground-based consistency results, then this bias needs to be removed.

Figure 25. Scatter distribution of SGRCM results for S-band weather radars in China.

#### 5. Conclusions


440

445

In conclusion, the two types of reflectivity factor comparison methods can be used as calibration methods for multi-band weather radars; however, there are also differences between them. Based on the analysis of precipitation events in 2024, it can

be observed that the bias range between S-band weather radar and surrounding radars of the same band is relatively large. The absolute bias from satellite-to-ground analysis is smaller than that from ground-based analysis, while the standard deviation from satellite-to-ground analysis is larger than that from ground-based analysis, indicating greater dispersion in the bias between satellite and ground-based radars. The two analytical methods for the X-band phased array weather radar show good consistency and both demonstrate the significant attenuation characteristic of the X-band phased array weather radar. Overall, the metrics from ground-based consistency analysis are greater than those from satellite-to-ground analysis, which may be due to the fact that the former considers data from the entire detection range, whereas the latter is limited by observation distance.

This distance limitation eliminates the impact caused by inconsistencies in radar beam pointing calibration in the overlapping distant observation regions. More observational samples are needed for further analysis of this effect.

#### **Appendix A: Weather Radar Hardware Information**

The relevant hardware information for the 19 S-band weather radars and 13 X-band phased array radars used in this study is summarized in the table below. For the "Operation mode" column, we use the following numerical codes:

- 1. All-solid-state amplification
- 2. Amplification chain where a solid-state amplifier drives a klystron amplifier
- 3. Active phased array

| ID | Band | Polarization type | Doppler    | Operation | Horizontal | Antenna  | Antenna   | Reflectivity |
|----|------|-------------------|------------|-----------|------------|----------|-----------|--------------|
|    |      |                   | processing | mode      | beamwidth  | diameter | gain (dB) | range        |
|    |      |                   | mode       |           | (°)        | (m)      |           |              |
| 1  | S    | dual polarization | FFT        | 1         | 0.972      | 8.5      | 45.17     | ≥460KM       |
| 2  | S    | dual polarization | FFT        | 2         | 0.99       | 8.54     | 44.5      | ≥460KM       |
| 3  | S    | dual polarization | FFT        | 1         | 0.92       | 8.5      | 45.62     | ≥460KM       |
| 4  | S    | dual polarization | FFT        | 1         | 0.98       | 8.5      | 44.7      | ≥460KM       |
| 5  | S    | dual polarization | FFT        | 2         | 0.94       | 8.534    | 45.1      | ≥460KM       |
| 6  | S    | dual polarization | FFT        | 2         | 0.981      | 8.5      | 45.96     | ≥460KM       |
| 7  | S    | dual polarization | FFT        | 2         | 0.912      | 8.45     | 45.11     | ≥460KM       |

| 8  | S | dual polarization | FFT | 2 | 0.967 | 8.54  | 45.12 | ≥460KM |
|----|---|-------------------|-----|---|-------|-------|-------|--------|
| 9  | S | dual polarization | FFT | 2 | 0.912 | 8.5   | 45.11 | ≥460KM |
| 10 | S | dual polarization | FFT | 2 | 0.927 | 8.54  | 45.49 | ≥460KM |
| 11 | S | dual polarization | FFT | 2 | 0.987 | 8.4   | 44.73 | ≥460KM |
| 12 | S | dual polarization | FFT | 2 | 0.9   | 8.534 | 45.8  | ≥460KM |
| 13 | S | dual polarization | FFT | 2 | 0.987 | 8.5   | 45.99 | ≥460KM |
| 14 | S | dual polarization | FFT | 2 | 0.97  | 8.5   | 45.6  | ≥230KM |
| 15 | S | dual polarization | FFT | 2 | 0.95  | 8.5   | 44.53 | ≥460KM |
| 16 | S | dual polarization | FFT | 1 | 0.916 | 8.534 | 45.54 | ≥460KM |
| 17 | S | dual polarization | FFT | 2 | 0.879 | 8.534 | 45.45 | ≥460KM |
| 18 | S | dual polarization | FFT | 2 | 0.97  | 8.5   | 44.79 | ≥460KM |
| 19 | S | dual polarization | FFT | 2 | 0.99  | 8.534 | 45.24 | ≥460KM |
| 20 | X | dual polarization | FFT | 3 | 0.972 | 0.7   | 36    | ≥230KM |
| 21 | X | dual polarization | FFT | 3 | 0.972 | 0.7   | 36    | ≥230KM |
| 22 | X | dual polarization | FFT | 3 | 0.972 | 0.7   | 36    | ≥230KM |
| 23 | X | dual polarization | FFT | 3 | 0.972 | 0.7   | 36    | ≥230KM |
| 24 | X | dual polarization | FFT | 3 | 0.972 | 0.7   | 36    | ≥230KM |
| 25 | X | dual polarization | FFT | 3 | 0.972 | 0.7   | 36    | ≥230KM |
| 26 | X | dual polarization | FFT | 3 | 0.972 | 0.7   | 36    | ≥230KM |
| 27 | X | dual polarization | FFT | 3 | 0.972 | 0.7   | 36    | ≥230KM |

| 28 | X | dual polarization | FFT | 3 | 0.972 | 0.7  | 36 | ≥230KM |
|----|---|-------------------|-----|---|-------|------|----|--------|
| 29 | X | dual polarization | FFT | 3 | 0.972 | 0.72 | 36 | ≥230KM |
| 30 | X | dual polarization | FFT | 3 | 0.972 | 0.72 | 36 | ≥230KM |
| 31 | X | dual polarization | FFT | 3 | 0.972 | 1.3  | 36 | ≥230KM |
| 32 | X | dual polarization | FFT | 3 | 0.972 | 0.72 | 36 | ≥230KM |

#### 470 Author Contributions

H.H.: Conceptualization, Methodology, Formal analysis, Writing—original draft preparation, Writing—review and editing, Supervision

S.P.: Conceptualization, Validation

L.W.: Data curation, Funding acquisition

475 N.S.: Project administration

C.Z.: Validation, Investigation H.W.: Investigation, Validation

Y.L.: Resources, Supervision

## **Competing interests**

The authors declare that they have no conflict of interest.

# Financial support

This research has been supported by the China Meteorological Service Association Meteorological Science and Technology Innovation Platform Project (No.: CMSA2023MB005) and the National Natural Science Foundation of China (No.: U2342216).

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
