# Peer review of "Design and Application of Two Consistency Verification Methods for Weather Radar Networks in the South China Region"

_EGUsphere, 2025_

## Community Comment (CC1)

This paper needs a major revision. I think the material is there, but a more thorough analysis is needed. A lot of information on the radars are missing (hardware signal processing e.gl) so that the results are hard to interpret. Figures are not properly discussed (e.g Figure 17) where two modes are visible. In principle you would expect one linear dependence (ideally 1:1).
**Reply:** Thank you for your valuable comments. We will revise the manuscript according to your suggestions. We will add information about the radar hardware and provide a more thorough discussion of the figures in the revised version.

Some more specific comments;

You    don't list other options to verify the calibration and consistency of data in the    network, most importantly the sun as a reference. Please include and discuss!
**Reply:** Thank you for your comment. Improving network consistency is a comprehensive task. The two inter-radar consistency analysis methods proposed in this paper represent the first stage of this work. The results provide a reference for operational staff, helping them identify which radars should be prioritized for calibration and correction in the second stage. During calibration, t Solar Calibration and Metal Sphere Calibration will be used as references to further identify and rectify specific radars. This paper mainly focuses on the method design and results analysis of the first stage.

A technical description of the radars you investigate is missing. This is important to interpret the results:
e.g. antenna gain, beam width,    transmit power, tx type, dualpol or not, signal processing, clutterfilter and so on.
**Reply:** This information was not confirmed for public disclosure before submission. We will verify which parts can be made public and include them in the revised manuscript after confirmation.

you missed https://amt.copernicus.org/preprints/amt-2021-257/amt-2021-257.pdf
check out this paper
**Reply:** Thank you for providing this information. We will carefully review it and include it in the references.

l 48: "satellite used as a reference standard": I would disagree here. No weather radar network is using satellite    radar data as a reference operationally
**Reply:** We fully agree with your point. The purpose of our study is to use the FY-3G satellite as a reference standard to analyze the deviations between ground-based radars, rather than to compare the differences between the satellite and radars themselves, as it is not possible to determine which observation is closest to the actual state of the target. Considering that FY-3G itself is calibrated for stability and consistency using other satellites such as GPM, we believe it can provide a stable long-term time series. In this sense, it serves as a reference standard among all the radars in the network.

l.100: is an attenuation correction performed with the Ku, Ka-Band data? or do you    avoid

situations with attenuation?

**Reply:** As mentioned in line 78 of the manuscript, the FY-3G Level 2 product was used, which contains reflectivity factor products for the S-band, C-band, and X-band. These reflectivity factors have been corrected for frequency, and I will provide references for the specific correction methods in the revised version of the manuscript.

1. 135: the specific mathematical detail to get the coordinates right, should be moved to an appendix, unless there is something new here.

**Reply:** Spatial matching is a key aspect for analyzing the observational consistency between adjacent radars. The formulas and methods presented are intended to better illustrate Figure 4. These technical details are also one of the innovative aspects of our approach, so we prefer to include them in the main text to facilitate readers' understanding and reproduction of the algorithm.

1. 232: is this a X-Band phased array? You don't use an attenuation correction?

this section needs be reworked in order to really provide a meaningful comparison between the two bands

**Reply:** The X-band weather radar used in line 232 of this paper is a all solid-state mechanical X-band radar, which is yet to be calibrated. We did not apply attenuation correction to its base data, and this part of the analysis is intended only to illustrate that, when comparing the consistency of radars operating at different bands, attenuation can cause variations in the results. In the subsequent analyses presented later in the paper, we use X-band phased array radars deployed in Guangdong Province, for which attenuation correction has already been applied to the base data. The specific correction method can be found in the following reference: Xiao LS, Hu DM, Chen S, et al., 2021. Study on attenuation correction algorithm of X-band dual polarization phased array radar [J]. Meteorological Monthly, 47(6): 703–716 (in Chinese). The methods described in Sections 3.1, 3.2, and 3.4 were actually used. We will add this reference in the revised version of the manuscript.

1. 244: the 15-35 dBZ: do you do an attenuation correction? or do you avoid any precipitation > 35 dBZ? but then 35 dBZ is probably too large for the X-Band; you will have attenuation. Please clarify.

**Reply:** As mentioned in the previous question, the reflectivity factors from the X-band phased array radars we used have undergone attenuation correction before generating the base data. When we limit the range of reflectivity factors, it is to focus our analysis on stable precipitation. For stronger convective precipitation, rapid changes in targets can lead to mismatches in the overlapping areas during comparison, where the targets may not correspond to the same echo. This approach helps to exclude errors caused by the weather process itself (rather than the radar system).

1. 12: I couldn't find a reference to figure 9. it is not clear which radar is the Radar1 or 2. Clearly state what radar is meant! what kind of correction is shown?

**Reply:** Thank you for your correction. The figure referenced in line 241 should be Figure 9. We will switch the positions of the left and right images according to common reader habits.

In Figure 9, we use the X-band radar as Radar 1 and the S-band radar as Radar 2. An adaptive attenuation correction method was used, and we will add a detailed description of this method and its references in the revised version of the manuscript.

l 265,  fig 10: no dualpol system? no sqi, Doppler filter implemented?
**Reply:** In Figure 10, the radar exhibiting sea clutter echoes is a dual-polarization radar. SQI is not involved in the signal processing; and one-dimensional and two-dimensional clutter Doppler filtering methods are applied.

1. 280: describe the fuzzy logic interference removal I think you mean the left figure as the quality controlled picture?

**Reply:** Yes, the left image is the quality-controlled one. We will adjust the order of the left and right images according to the readers' reading habits. The identification and removal of radial interference echoes were mainly performed using a fuzzy logic method. Four characteristic parameters reflecting the differences between radial interference echoes and precipitation echoes were extracted from the reflectivity factor, including:

RREF, representing the continuity of the reflectivity factor along the current radial (as shown in Equations (1)-(2));

dZ, indicating the consistency of echo power in the adjacent range bins along the current radial (as shown in Equations (3)-( 5));

TDBZ (unit: dB²), expressing the local textural consistency of reflectivity along the radial (as shown in Equation (6));

SPIN, representing the sign changes of adjacent reflectivity factors within a local area (as shown in Equations (7)-(8)).

$$R_{\mathrm{REF}} = \frac{\sum_{i=0}^{N_{\mathrm{R}}} N_{\mathrm{Z}}}{N_{\mathrm{R}}} \times 100\% \tag{1}$$

$$N_{\mathrm{Z}} = \begin{cases} 1 & Z_{i,j} = Val \\ 0 & Z_{i,j} \neq Val \end{cases} \tag{2}$$

$$B_{i,j} = Z_{i,j} - 20\lg R_{i,j} \tag{3}$$

$$\overline{B} = \frac{\sum_{N_{\mathrm{R}*0.9}}^{N_{\mathrm{R}}} B_{i,j}}{N_{\mathrm{R}} * 0.1} \tag{4}$$

$$\mathrm{dZ} = B_{i,j} - \overline{B} \tag{5}$$

$$T_{\mathrm{DBZ}} = \frac{\sum_{j=-5}^{j=5} (Z_{i,j} - Z_{i,j+1})^2}{11} \tag{6}$$

$$M_{S_{\mathrm{PIN}}} = \begin{cases} 1 & \dfrac{|Z_{i,j} - Z_{i,j-1}| + |Z_{i,j+1} - Z_{i,j}|}{2} > Z_{\mathrm{thresh}} \\ 0 & \dfrac{|Z_{i,j} - Z_{i,j-1}| + |Z_{i,j+1} - Z_{i,j}|}{2} \leq Z_{\mathrm{thresh}} \end{cases} \tag{7}$$

$$S_{\text{PIN}} = \sum_{j=-5}^{j=5} M_{S_{\text{PIN}}} \qquad (8)$$

In the equations, $Zi,j$ (unit: dBZ) is the reflectivity factor at a certain range bin, $Val$ is the effective detection value (unit: dBZ), $Ri,j$ is the distance between the range bin and the radar (unit: km), $NR$ is the number of range bins for the reflectivity factor, and $Z_{\text{thresh}}Z_{thresh}$ is the threshold for changes in the reflectivity factor between range bins.

For specific technical details, please refer to the following literature: Wen Hao, Zhang Lejian, Liang Haihe, Zhang Yang. 2020. "Radial interference echo identification algorithm based on fuzzy logic for weather radar." Acta Meteorologica Sinica, 78(1): 116-127. We will add this reference in the revised version of the manuscript.

l 310: figure 15:  I don't understand this figure. How does the ground based consistence analysis looks like? Take radar 1: what is the reference radar here? How do you come up with the bias?

**Reply:** When analyzing ground-based consistency, we set a distance threshold between adjacent radars (for example, 200 km between S-band radars), so any two radars within this threshold can be paired for matching. For Radar 1, if it can be paired with five surrounding radars, we calculate the bias between Radar 1 and each matched radar for every volume scan according to the method described in the paper. After one volume scan, Radar 1 will have five comparison results, and we take the mean of these five results as the final result for Radar 1 at that time. In this way, if Radar 1 has a significant systematic bias, it will be reflected in the bias result. If the standard deviation is large, it indicates that the observations from this radar are more dispersed and that further calibration and detailed analysis of the hardware are necessary.

Fig 16: font cannot be read. Rework the figures.  X-Axis is a time axis. What time period? why not showing the times? Larger biases can be attributed to specific weather events? Are there any snow cases?

**Reply:** We will redraw these figures in the revised version of the paper, increasing the font size for better readability. The X-axis represents the number of samples, with each sample corresponding to one volume scan. Since not every volume scan contains precipitation and meets the algorithm's threshold requirements, using time as the X-axis would result in discontinuity, so we used the sample count instead. Larger biases are closely related to specific weather events; convective weather, in particular, tends to produce larger biases due to the rapid movement and variability of targets. Additionally, because our analysis focuses on the southern coastal region, snowfall cases are expected to be very rare.

l319:  reflectivity is not "strong" it is large, small I would say

**Reply:** Thank you for your suggestion. We will address this issue in the revised version of the manuscript.

Fig 17: clearly two modes are visible in each plot, they are  not discussed and explained!  (two linear fits with different slopes could be fitted). Two modes suggest that

there is something fundamentally wrong, or?

**Reply:** We will add a discussion of the two modes in the revised version of the paper. As long as there is a precipitation event, ground-based consistency analysis will produce comparison results. However, since it takes FY-3G about 1–2 days to pass over the same location, and its spatial resolution is lower than that of radar, there is a significant difference in both the number of samples and the temporal frequency between the two methods. Therefore, for now, we have not considered analyzing the results of the two modes together in the same figure.

l. 328: without discussing the quality control of the reflectivity factor from the X-Band the results are difficult to interpret: are you really sure that you can rule out attenuation effects e.g.?

**Reply:** We used X-band phased array radars distributed in Guangdong Province, and attenuation correction algorithm has already been applied prior to the generation of the base data.

l 335: so Fig 19: really doesn't say anything about the biases. Comparing Fig 19 and 15 one would assume similar performance of the S and X-Band. Why do you show standard deviations? Doesn't make sense to me. Please explain!

**Reply:** In our radar network consistency analysis, we calculated several metrics, including bias, standard deviation, and correlation coefficient. The standard deviation reflects the dispersion of reflectivity bias as well as the stability of system observations. In Figure 15, the standard deviation of ground-based S-band weather radar consistency refers to the results of comparisons between S-band radars. Figure 19 shows the standard deviation between X-band phased array radars, which only reflects the dispersion of observational bias among radars with the same band and system. Factors such as the distance between overlapping areas and weather processes also have an impact. Of course, the final assessment of consistency is still primarily based on bias.

l 361: what is a SC model weather radar?

**Reply:** The SC radar is a model of S-band weather radar that operates within the operational weather radar network.

Fig 23: the result suggests that the satellite / radar has further systematic problems errors in my view. The calibration does not provide a more consistent result.

**Reply:** Thank you for your comment. Figure 23 shows that the bias between the satellite and radar has always existed. The smaller bias on the right side may be due to the limited number of observed targets in the 30–35 dBZ range. Since satellite observations have relatively low temporal and spatial frequency, we will collect more weather events for further analysis.

---

## Author Response (AR1)

**Response to review comments-1**

1. On pages 85 to 90 of the article, the VCP21 scanning mode is mainly used for stratiform precipitation, while VCP31 is primarily used for clear-sky conditions. It is recommended to make the changes in the text after confirmation.

**Reply:** We have confirmed the definition of VCP and made corrections in lines 93–94 of the revised version.

- 2. Section 3.1.2 mentions the removal of shielding caused by terrain. However, in recent years, shielding caused by buildings at low elevation angles has become increasingly common. How is this factor considered in the algorithm to ensure the reliability of the final results? **Reply:** Presently, we have incorporated configurable parameters within the algorithm. For stations subject to significant obstruction—including those caused by buildings—the lowest elevation angle (0.5°) is excluded from comparative analyses. Additionally, we are developing an algorithm to systematically assess the actual shielding conditions at each station. Once refined, this will be integrated into the model to enhance overall reliability.
- 3. How can the stability and accuracy of satellite data be ensured, and how can the results of satellite-ground comparisons be used to calibrate the radar?

**Reply:** FY-3G will be cross-compared with other satellites such as GPM to ensure a robust and consistent calibration mechanism. Within this framework, we propose using FY-3G as a reference benchmark to evaluate the consistency among ground-based radar networks, rather than directly calibrating discrepancies between radar and satellite observations. This approach is adopted due to the inherent difficulty in establishing which observation—satellite or radar—more accurately represents the true characteristics of the observed targets.

**Response to review comments-2**

This paper needs a major revision. I think the material is there, but a more thorough analysis is needed. A lot of information on the radars are missing (hardware signal processing e.gl) so that the results are hard to interpret. Figures are not properly discussed (e.g Figure 17) where two modes are visible. In principle you would expect one linear dependence (ideally 1:1).

**Reply:** Thank you for your valuable comments. We will revise the manuscript according to your suggestions. We will add information about the radar hardware and provide a more thorough discussion of the figures in the revised version.

**Changes in manuscript:** We have revised the abstract of the paper and added detailed descriptions of the two methods in Sections 2.2.1 and 2.2.2, including the algorithm workflows and the specific parameters used. We have compiled the radar hardware parameters as Appendix A and have revised some of the figure captions. Please refer to the revised version of the manuscript.

Some more specific comments;

**1.**You don't list other options to verify the calibration and consistency of data in the network, most importantly the sun as a reference. Please include and discuss!

**Reply:** Thank you for your comment. Improving network consistency is a comprehensive task. The two inter-radar consistency analysis methods proposed in this paper represent the first stage of this work. The results provide a reference for operational staff, helping them identify which radars should be prioritized for calibration and correction in the second stage. During calibration, t Solar Calibration and Metal Sphere Calibration will be used as references to further identify and rectify specific radars. This paper mainly focuses on the method design and results analysis of the first stage.

**Changes in manuscript:** We have added relevant descriptions in lines 434–437 of the discussion section in the revised version and have included additional references.

**2.**A technical description of the radars you investigate is missing. This is important to interpret the results:

e.g. antenna gain, beam width, transmit power, tx type, dualpol or not, signal processing, clutterfilter and so on.

**Reply:** This information was not confirmed for public disclosure before submission. We will verify which parts can be made public and include them in the revised manuscript after confirmation.

Changes in manuscript: We have compiled the radar hardware parameters as Appendix A.

**3.**you missed https://amt.copernicus.org/preprints/amt-2021-257/amt-2021-257.pdf check out this paper

**Reply:** Thank you for providing this information. We will carefully review it and include it in the references.

Changes in manuscript: We have added this reference in line 55 of the revised version.

**4.**I 48: "satellite used as a reference standard": I would disagree here. No weather radar network is using satellite radar data as a reference operationally

**Reply:** We fully agree with your point. The purpose of our study is to use the FY-3G satellite as a reference standard to analyze the deviations between ground-based radars, rather than to compare the differences between the satellite and radars themselves, as it is not possible to determine which observation is closest to the actual state of the target. Considering that FY-3G itself is calibrated for stability and consistency using other satellites such as GPM, we believe it can provide a stable long-term time series. In this sense, it serves as a reference standard among all the radars in the network.

**5.**I.100: is an attenuation correction performed with the Ku, Ka-Band data? or do you avoid situations with attenuation?

**Reply:** As mentioned in line 78 of the manuscript, the FY-3G Level 2 product was used, which contains reflectivity factor products for the S-band, C-band, and X-band. These reflectivity factors have been corrected for frequency, and I will provide references for the specific correction methods in the revised version of the manuscript.

Changes in manuscript: We have added the reference in line 82 of the revised version. Please

refer to this reference for the specific correction method.

**6.**135: the specific mathematical detail to get the coordinates right, should be moved to an appendix, unless there is something new here.

**Reply:** Spatial matching is a key aspect for analyzing the observational consistency between adjacent radars. The formulas and methods presented are intended to better illustrate Figure 4. These technical details are also one of the innovative aspects of our approach, so we prefer to include them in the main text to facilitate readers' understanding and reproduction of the algorithm.

**7.**232: is this a X-Band phased array? You don't use an attenuation correction? this section needs be reworked in order to really provide a meaningful comparison between the two bands

Reply: The X-band weather radar used in line 232 of this paper is a all solid-state mechanical X-band radar, which is yet to be calibrated. We did not apply attenuation correction to its base data, and this part of the analysis is intended only to illustrate that, when comparing the consistency of radars operating at different bands, attenuation can cause variations in the results. In the subsequent analyses presented later in the paper, we use X-band phased array radars deployed in Guangdong Province, for which attenuation correction has already been applied to the base data. The specific correction method can be found in the following reference: Xiao LS, Hu DM, Chen S, et al., 2021. Study on attenuation correction algorithm of X-band dual polarization phased array radar [J]. Meteorological Monthly, 47(6): 703–716 (in Chinese). The methods described in Sections 3.1, 3.2, and 3.4 were actually used. We will add this reference in the revised version of the manuscript.

**Changes in manuscript:** We have added the reference in line 297 of the revised version. Please refer to this reference for the specific correction method.

**8.**244: the 15-35 dBZ: do you do an attenuation correction? or do you avoid any precipitation > 35 dBZ? but then 35 dBZ is probably too large for the X-Band; you will have attenuation. Please clarify.

**Reply:** As mentioned in the previous question, the reflectivity factors from the X-band phased array radars we used have undergone attenuation correction before generating the base data. When we limit the range of reflectivity factors, it is to focus our analysis on stable precipitation. For stronger convective precipitation, rapid changes in targets can lead to mismatches in the overlapping areas during comparison, where the targets may not correspond to the same echo. This approach helps to exclude errors caused by the weather process itself (rather than the radar system).

Changes in manuscript: We have added a reference in line 169 of the revised manuscript.

**9.**12: I couldn't find a reference to figure 9. it is not clear which radar is the Radar1 or 2. Clearly state what radar is meant! what kind of correction is shown?

**Reply:** Thank you for your correction. The figure referenced in line 241 should be Figure 9. We will switch the positions of the left and right images according to common reader habits. In Figure 9, we use the X-band radar as Radar 1 and the S-band radar as Radar 2. An adaptive

attenuation correction method was used, and we will add a detailed description of this method and its references in the revised version of the manuscript.

**Changes in manuscript:** We have revised the numbering of Figure 9 and cited it in line 295. In line 297 of the revised version, we have added the reference; for the adaptive attenuation correction method, please refer to this reference. The description of the radar has been included in the caption of Figure 10.

10.1 265, fig 10: no dualpol system? no sqi, Doppler filter implemented?

**Reply:** In Figure 10, the radar exhibiting sea clutter echoes is a dual-polarization radar. SQI is not involved in the signal processing; and one-dimensional and two-dimensional clutter Doppler filtering methods are applied.

**11.**280: describe the fuzzy logic interference removal I think you mean the left figure as the quality controlled picture?

**Reply:** Yes, the left image is the quality-controlled one. We will adjust the order of the left and right images according to the readers' reading habits. The identification and removal of radial interference echoes were mainly performed using a fuzzy logic method. Four characteristic parameters reflecting the differences between radial interference echoes and precipitation echoes were extracted from the reflectivity factor, including:

RREF, representing the continuity of the reflectivity factor along the current radial (as shown in Equations (1)-(2));

dZ, indicating the consistency of echo power in the adjacent range bins along the current radial (as shown in Equations (3)-(5));

TDBZ (unit: dB²), expressing the local textural consistency of reflectivity along the radial (as shown in Equation (6));

SPIN, representing the sign changes of adjacent reflectivity factors within a local area (as shown in Equations (7)-(8)).

$$R_{\rm REF} = \frac{\sum_{i=0}^{N_{\rm R}} N_{\rm Z}}{N_{\rm R}} \times 100\% \tag{1}$$

$$N_{\rm Z} = \begin{cases} 1 & Z_{i,j} = Val \\ 0 & Z_{i,j} \neq Val \end{cases} \tag{2}$$

$$B_{i,j} = Z_{i,j} - 20 \lg R_{i,j} \tag{3}$$

$$\overline{B} = \frac{\sum_{N_{R*0.9}}^{N_R} B_{i,j}}{N_R * 0.1} \tag{4}$$

$$dZ = B_{i,j} - \overline{B} \tag{5}$$

$$T_{\text{DBZ}} = \frac{\sum_{j=-5}^{j=5} (Z_{i,j} - Z_{i,j+1})^2}{11}$$
 (6)

$$M_{S_{\text{PIN}}} = \begin{cases} 1 & \frac{\left|Z_{i,j} - Z_{i,j-1}\right| + \left|Z_{i,j+1} - Z_{i,j}\right|}{2} > Z_{\text{thresh}} \\ 0 & \frac{\left|Z_{i,j} - Z_{i,j-1}\right| + \left|Z_{i,j+1} - Z_{i,j}\right|}{2} \le Z_{\text{thresh}} \end{cases}$$
(7)

$$S_{\text{PIN}} = \sum_{i=-5}^{j=5} M_{S_{\text{PIN}}} \tag{8}$$

In the equations, Zij (unit: dBZ) is the reflectivity factor at a certain range bin, Val is the effective detection value (unit: dBZ), Rij is the distance between the range bin and the radar (unit: km), NR is the number of range bins for the reflectivity factor, and  $Z_{thresh}Z_{thresh}$  is the threshold for changes in the reflectivity factor between range bins.

For specific technical details, please refer to the following literature: Wen Hao, Zhang Lejian, Liang Haihe, Zhang Yang. 2020. "Radial interference echo identification algorithm based on fuzzy logic for weather radar." Acta Meteorologica Sinica, 78(1): 116-127. We will add this reference in the revised version of the manuscript.

**Changes in manuscript:** We have added the reference in line 341 of the revised version. Please refer to this reference for the specific interference removal method.

**12.**I 310: figure 15: I don't understand this figure. How does the ground based consistence analysis looks like? Take radar 1: what is the reference radar here? How do you come up with the bias?

**Reply:** When analyzing ground-based consistency, we set a distance threshold between adjacent radars (for example, 200 km between S-band radars), so any two radars within this threshold can be paired for matching. For Radar 1, if it can be paired with five surrounding radars, we calculate the bias between Radar 1 and each matched radar for every volume scan according to the method described in the paper. After one volume scan, Radar 1 will have five comparison results, and we take the mean of these five results as the final result for Radar 1 at that time. In this way, if Radar 1 has a significant systematic bias, it will be reflected in the bias result. If the standard deviation is large, it indicates that the observations from this radar are more dispersed and that further calibration and detailed analysis of the hardware are necessary.

**Changes in manuscript:** We have added a description of the method in lines 357–362 of the revised version.

**13.**Fig 16: font cannot be read. Rework the figures. X-Axis is a time axis. What time period? why not showing the times? Larger biases can be attributed to specific weather events? Are there any snow cases?

**Reply:** We will redraw these figures in the revised version of the paper, increasing the font size for better readability. The X-axis represents the number of samples, with each sample corresponding to one volume scan. Since not every volume scan contains precipitation and meets the algorithm's threshold requirements, using time as the X-axis would result in discontinuity, so we used the sample count instead. Larger biases are closely related to specific weather events; convective weather, in particular, tends to produce larger biases due to the rapid movement and variability of targets. Additionally, because our analysis focuses on the

southern coastal region, snowfall cases are expected to be very rare.

**Changes in manuscript:** We have numbered the radars in Appendix A and redrawn the figure accordingly; please see Figure 17 in the revised version.

**14.**I319: reflectivity is not "strong" it is large, small I would say

**Reply:** Thank you for your suggestion. We will address this issue in the revised version of the manuscript.

Changes in manuscript: We have made corrections in line 403 of the revised version.

**15.**Fig 17: clearly two modes are visible in each plot, they are not discussed and explained! (two linear fits with different slopes could be fitted). Two modes suggest that there is something fundamentally wrong, or?

**Reply:** We will add a discussion of the two modes in the revised version of the paper. As long as there is a precipitation event, ground-based consistency analysis will produce comparison results. However, since it takes FY-3G about 1–2 days to pass over the same location, and its spatial resolution is lower than that of radar, there is a significant difference in both the number of samples and the temporal frequency between the two methods. Therefore, for now, we have not considered analyzing the results of the two modes together in the same figure.

**16.**I. 328: without discussing the quality control of the reflectivity factor from the X-Band the results are difficult to interpret: are you really sure that you can rule out attenuation effects e.g.?

**Reply:** We used X-band phased array radars distributed in Guangdong Province, and attenuation correction algorithm has already been applied prior to the generation of the base data.

**Changes in manuscript:** We have added the description and the reference in line 398 of the revised version.

**17**.I 335: so Fig 19: really doesn't say anything about the biases. Comparing Fig 19 and 15 one would assume similar performance of the S and X-Band. Why do you show standard deviations? Doesn't make sense to me. Please explain!

**Reply:** In our radar network consistency analysis, we calculated several metrics, including bias, standard deviation, and correlation coefficient. The standard deviation reflects the dispersion of reflectivity bias as well as the stability of system observations. In Figure 15, the standard deviation of ground-based S-band weather radar consistency refers to the results of comparisons between S-band radars. Figure 19 shows the standard deviation between X-band phased array radars, which only reflects the dispersion of observational bias among radars with the same band and system. Factors such as the distance between overlapping areas and weather processes also have an impact. Of course, the final assessment of consistency is still primarily based on bias.

18.1 361: what is a SC model weather radar?

Reply: The SC radar is a model of S-band weather radar that operates within the operational

weather radar network.

Changes in manuscript: We have provided an explanation in line 438 of the revised version.

**19.**Fig 23: the result suggests that the satellite / radar has further systematic problems errors in my view. The calibration does not provide a more consistent result.

**Reply:** Thank you for your comment. Figure 23 shows that the bias between the satellite and radar has always existed. The smaller bias on the right side may be due to the limited number of observed targets in the 30–35 dBZ range. Since satellite observations have relatively low temporal and spatial frequency, we will collect more weather events for further analysis.

**Response to review comments-3**

This manuscript proposes methods for verification of weather radar networks. Not only by ground-based radars, but also space-borme radars are used for consistency verification. I have several critical queries to be solved before the final decision.

**General Comments:**

• The overall direction and purpose of the manuscript remain unclear. Additionally, the description of the analysis methodology is insufficient, making it impossible to reproduce the results based on the current manuscript.

**Reply:** Thank you for your comments. In the revised version of the manuscript, we will clarify the research objectives, provide a more detailed description of the methodology, include a methodological framework, and specify the parameters used. These additions will help readers to better reproduce the algorithms and results presented in the paper.

**Changes in manuscript:** We have revised the abstract of the paper and added detailed descriptions of the two methods in Sections 2.2.1 and 2.2.2, including the algorithm workflows and the specific parameters used.

• Despite the abundance of radar systems in China, the authors do not specify which radars or what time periods were used in the analysis. Furthermore, the text-only explanation of the radar locations is difficult to interpret. At minimum, a map of the radar network should be included to facilitate understanding.

**Reply:** In the revised version of the manuscript, we will add descriptions of the radar hardware and the analysis period, and we will include a map showing the locations of the radar network. **Changes in manuscript:** We have added a site distribution map as Figure 2.

• The study investigates biases through comparisons between ground-based radars and between ground-based and satellite radars. However, such comparisons merely highlight the relative biases between systems, and an independent, well-calibrated reference radar is essential. Is there no such calibrated radar within the network used in this study?

**Reply:** In practical work, we have established a radar calibration center in Changsha, Hunan, where an S-band dual-polarization radar undergoes regular calibration and serves as the reference radar in ground-satellite comparison experiments. When analyzing the consistency

of the ground-based radar network, twenty reference radars across the country (including the one in Changsha) are selected to analyze biases. This work has just started this year, and the specific selection criteria and methods are still being refined.

It is also unclear what types of biases the authors are attempting to identify. Are these parameters that cannot be corrected through individual radar calibration, or are they related to factors like beam blockage or system biases that can be corrected? The manuscript lacks clarity on this point. Additionally, even if biases are identified, the manuscript does not explain how this information will be used—whether for correction or simply as observational insight. Reply: In the analysis of radar network consistency, three metrics are used: bias, standard deviation, and correlation coefficient. Bias reflects the systematic deviations between radars, standard deviation indicates the dispersion of radar observations, and the correlation coefficient is greatly influenced by sample size, so it has not been analyzed at this stage. In practical work, once the method designed in this paper indicates that a radar exhibits bias, we conduct detailed calibration procedures, including tests for beam pointing, antenna performance parameters, transmitter output pulse width and peak power, and feeder loss. Additionally, we use methods such as Solar Calibration and Metal Sphere Calibration for verification. Issues that may be discovered include the CW output signal of the frequency source being lower than the originally recorded value, uncalibrated azimuth after radar maintenance, and errors in measuring radome/transceiver feeder losses. This paper mainly introduces a radar network consistency analysis method designed based on raw data, which serves primarily as an indication. Only after thorough calibration can the root causes be truly traced and rectification suggestions proposed. We will further elaborate on this in the revised version of the manuscript.

**Changes in manuscript:** We have added relevant descriptions in the abstract, Section 3.2.1, and the discussion in Section 4.

• Although the term "model" is used, the methodology appears to be more of a data extraction and comparison approach rather than a model in the conventional sense.

**Reply:** Thank you for your suggestion. In the revised version of the manuscript, we will replace the term "model" with "method".

**Changes in manuscript:** We have replaced the term "model" with "method".

• The manuscript refers to numerous parameters used in data extraction, but they are scattered throughout the text and difficult to follow. Parameters such as thresholds should be clearly summarized in a table.

**Reply:** We will add a description of the methodological workflow and parameter thresholds to the methods section.

**Changes in manuscript:** We have added detailed descriptions of the two methods in Sections 2.2.1 and 2.2.2, including the algorithm workflows and the specific parameters used.

• From Section 3.1.3 onward, the statistical analyses lack clarity regarding which radar(s) and what data periods were used. Without a clear listing of these, the reliability and reproducibility of the analysis cannot be ensured.

**Reply:** We will supplement the manuscript with details on the radar hardware parameters and data periods in the revised version.

**Changes in manuscript:** We have compiled the radar hardware parameters in Appendix A. The data period used is explained in line 362.

• There is insufficient explanation of the analysis methods. For example, in the paragraph starting on P.10 L.219, how was VIL calculated? Also, which radar stations correspond to Radar1 and Radar2 in Figure 7?

**Reply:** We will add a description of the VIL calculation method in the revised version of the manuscript and clearly specify the names of the radars used in the paper.

**Changes in manuscript:** We have added a reference for the VIL calculation method, and included the radar ID information between lines 276 and 277.

• In Section 3.2 and onward, only a subset of the presumably large dataset is shown. However, since the selection criteria are not explained, the reliability of the results is questionable—for example, in Figure 20.

**Reply:** During the comparison, we limit the range of reflectivity factor and signal-to-noise ratio to exclude the effects of rapidly changing convective precipitation and weak signals. We will provide a detailed description of the data selection criteria in the revised version of the manuscript.

**Changes in manuscript:** We have revised the abstract of the paper and added detailed descriptions of the two methods in Sections 2.2.1 and 2.2.2, including the algorithm workflows and the specific parameters used.

**Specific Comments:**

• P.3 L.78: The phrase "corrected for frequency" is unclear, as reflectivity in the Rayleigh scattering regime is not wavelength-dependent. Please clarify what correction was applied and how.

**Reply:** As mentioned in line 78 of the manuscript, the FY-3G Level 2 product was used, which contains reflectivity factor products for the S-band, C-band, and X-band. These reflectivity factors have been corrected for frequency, Please refer to the following reference for the specific method used:

Wu Qiong, Yang Meilin, Chen Lin, Yin Honggang, Shang Jian, Gu Songyan. 2023. A frequency correction algorithm for spaceborne precipitation measurement radar and ground-based weather radar. Acta Meteorologica Sinica, 81(2): 353–360.

**Changes in manuscript:** We have added references for the specific correction methods in the revised version of the manuscript, line 82.

• P.3 L.87: Please write out "VCP" (Volume Coverage Pattern) in full upon first use.

**Reply:** We will include this in the revised version of the manuscript.

**Changes in manuscript:** We have included this in line 93.

P.3 Figure 1: Indicate the satellite's direction of movement directly on the figure.

Reply: We will make this revision in the revised version of the manuscript.

**Changes in manuscript:** We have added the satellite's direction of movement in the annotation of Figure 1.

• P.4 L.88: The phrase "Evaluation results from 2024...in this study" requires a citation.

**Reply:** The evaluation mentioned in line 88 refers to routine assessments conducted as part of operational work, and no related papers have been published. We selected several stations to analyze and compare the consistency of ground-based observations before and after the mode switching. The change in reflectivity deviation between the two observation modes before and after mode switching is relatively small, within  $\pm 0.4$  dB, indicating that the radar reflectivity remains quite consistent before and after the mode switch. The following table provides examples from the analysis.

| Station ID | Model | Mode  | Time            | Bias(dB)
VCP11-
VCP21 |
|------------|-------|-------|-----------------|-----------------------------|
| Z9371      | SAD   | VCP11 | 2024/6/28 21:00 | 0.17                        |
|            |       | VCP21 | 2024/6/30 20:00 |                             |
| Z9376      | SAD   | VCP11 | 2024/7/1 23:00  | -0.22                       |
|            |       | VCP21 | 2024/7/1 22:00  |                             |
| Z9377      | SB    | VCP11 | 2024/7/1 16:00  | 0.2                         |
|            |       | VCP21 | 2024/7/1 19:00  |                             |
| Z9379      | SAD   | VCP11 | 2024/7/1 21:00  | -0.11                       |
|            |       | VCP21 | 2024/7/1 23:00  |                             |
| Z9396      | SB    | VCP11 | 2024/7/2 0:00   | -0.04                       |
|            |       | VCP21 | 2024/7/2 1:00   |                             |
| Z9551      | SA    | VCP11 | 2024/6/27 2:00  | 0.1                         |
|            |       | VCP21 | 2024/6/27 4:00  |                             |
| Z9552      | SAD   | VCP11 | 2024/6/28 18:00 | -0.19                       |
|            |       | VCP21 | 2024/6/28 19:00 |                             |
| Z9555      | CCD   | VCP11 | 2024/6/23 4:00  | -0.14                       |
|            |       | VCP21 | 2024/6/23 7:00  |                             |
| Z9556      | SAD   | VCP11 | 2024/7/2 7:00   | -0.09                       |
|            |       | VCP21 | 2024/7/2 9:00   |                             |
| Z9562      | SAD   | VCP11 | 2024/6/25 17:00 | -0.4                        |
|            |       | VCP21 | 2024/6/25 18:00 |                             |

**Changes in manuscript:** We have added supplementary explanations in lines 94–96 of the revised manuscript.

• P.4 L.96–97: The terms "PRE" and "FRE" are undefined and should be explained.

**Reply:** The "PRE" and "FRE" mentioned in lines 96-97 refer to the data preprocessing module and the frequency correction module, respectively. We will include this clarification in the revised version of the paper.

Changes in manuscript: We have added an explanation in lines 113–114.

• P.4 L.102: It is unclear what the "first and second reference frames" refer to.

**Reply:** The "first and second reference frames" mentioned in line 102 are indeed based on relevant literature. We will provide further explanation and add the appropriate references in the revised manuscript.

Yang Hongping, Han Wei, Wang Hui, Heng Hu, A Reference Positioning Methodology for Computing GeodeticCoordinates of Radar Echo, Meteorological Science and Technology,2023,51(1):22-30.(In Chinese)

Changes in manuscript: We have added a reference in line 120 of the revised manuscript.

• **P.4 L.110**: Explain how the averaging and gridding were performed. These procedures can introduce bias and should be described in detail.

**Reply:** The steps for satellite–ground consistency comparison are as follows:

(1) Spatial and Temporal Collocation

Begin by identifying ground-based radars (GB) whose observational coverage significantly overlaps with the FY-3G PMR (SG) scanning region. Overlap criteria require that at least 3,000 (S/C-band) or 400 (X-band) PMR grid points fall within the GB's observation area. For temporal alignment, only data pairs where the observation times differ by less than 180 seconds are retained.

**(2) Resampling**

The FY-3G PMR Ku L2 product is a resampling dataset with 400 bins and a vertical resolution of 50 m, which differs from the original vertical resolution of 250 m used in the SG scanning mode. In this study, the data at each scanning track grid of SG are resampled into a four-dimensional (longitude, latitude, height, time) grid data with a vertical resolution of 250 m (80 bins) and a horizontal track resolution of 5 km, as the SG scanning mode. That is, each SG grid is 5 km  $\times$  5 km  $\times$  250 m. Measurements that are too close to or too far away from the GB stations have significant errors. Through multiple experiments, this study selects the time-paired GB reflectivity data with a distance of 50-150 km away from the stations for S/C-band GBs and 9-42 km for X-band GBs. The GB reflectivity data are then transformed into three-dimensional (longitude, latitude, height) data.

**(3) Extraction of Stratiform Rain Cases**

Stratiform precipitation is isolated using the precipitation classification provided by the SG product at each grid point. Both satellite and ground-based reflectivity values are further restricted to 20–35 dBZ within the 2–4 km altitude range to focus on relatively stable echoes.

(4) Pairwise Data Construction

For each spatial-temporal matchup, if multiple GB range bins correspond to a single SG grid cell, they are averaged to produce a composite GB reflectivity value. These paired values—SG and averaged GB reflectivity—form the basis for subsequent comparison.

(5) Consistency Assessment

When at least 20 such matched pairs are available, key statistical indices—namely bias, standard deviation, and correlation coefficient—are computed to quantitatively evaluate the consistency between the SG and the GB network. We will include this in the revised version of the manuscript.

**Changes in manuscript:** We have added detailed calculation procedures and parameter thresholds in lines 120–146 of the revised manuscript.

• P.6 L.120: The term "S-PAR" is undefined and should be clarified.

**Reply:** S-PAR refers to S-band phased array radar. We will add this explanation in the revised version of the manuscript.

Changes in manuscript: We have made a correction in line 178 of the revised manuscript.

• **P.7 L.138**: The meaning of "Km = 4/3" is unclear and should be explained.

Reply: In meteorology and radar meteorology, Km=4/3 usually refers to the Effective Earth

Radius Factor. We will add this explanation in the revised version of the manuscript.

**Changes in manuscript:** We have provided an explanation in line 193 of the revised manuscript.

• **P.7 L.160**: The variable "Hthre" should be written with a subscript for clarity.

**Reply:** We will make this correction in the revised version of the manuscript.

**Changes in manuscript:** We have made a correction in lines 215–216 of the revised manuscript.

• **P.8 L.170**: If comparing a single satellite with a single ground radar, vertical resolution should not be an issue. The intent of this sentence is unclear. If multiple ground radars are being matched to one satellite, this should be clearly stated.

**Reply:** The description in the article is unclear. A single FY-3G PMR grid cell may contain one or more ground-based radar range bins. The reflectivity values of these range bins are averaged to obtain a new ground-based reflectivity value that corresponds to the FY-3G PMR grid cell reflectivity. The resulting pair of the FY-3G PMR grid cell reflectivity and the new ground-based reflectivity constitutes a comparison pair. By performing this spatial and temporal matching for multiple FY-3G PMR grid cells, a set of comparison pairs is formed.

**Changes in manuscript:** We have added detailed calculation procedures and parameter thresholds in lines 120–146 of the revised manuscript.

• **P.11 L.244**: Justification is needed for choosing the reflectivity range of 15–35 dBZ. If rain attenuation is a concern, then strong reflectivity along the beam path should also be considered for exclusion. Please elaborate.

**Reply:** The range of 15–35 dBZ was chosen to retain stable stratiform precipitation echoes. Convective precipitation echoes with stronger reflectivity tend to vary rapidly, making it difficult to ensure that different radars are observing the same echo region.

**Changes in manuscript:** We have added a reference in line 169 of the revised manuscript.

• **P.11 Figure 8**: It is unclear which result corresponds to the S-band radar.

**Reply:** The image on the right in Figure 8 corresponds to the S-band radar. We will add this label in the revised version of the manuscript.

**Changes in manuscript:** We have clarified the explanation of the radar in the caption of Figure 8.

• **P.13 Figure 10**: The relative positions of the radars are not shown. Without this context, comparing the two radars is impossible for readers. At minimum, the coastline should be shown, and the map axes (latitude/longitude) should be consistent across both subplots.

**Reply:** We will revise this figure in the updated version of the manuscript.

**Changes in manuscript:** We have added the coastline display in the figure. Please refer to the revised Figure 11.

• **P.13 Figure 11**: The figure does not indicate what parameter is being visualized. Please clarify.

**Reply:** The right panel in Figure 11 shows the calculated Ref SD values. Each of the first four range rings outward from the radar station represents 100 kilometers, and the outermost range ring represents 460 kilometers. We will add this explanation in the revised version of the manuscript.

**Changes in manuscript:** We have added this explanation in the revised version. Please refer to Figure 12.

• **P.14 L.280**: The term "Fuzzy logic" is mentioned without describing the actual algorithm or implementation used.

**Reply:** The identification and removal of radial interference echoes were mainly performed using a fuzzy logic method. Four characteristic parameters reflecting the differences between radial interference echoes and precipitation echoes were extracted from the reflectivity factor, including:

**RREF,** representing the continuity of the reflectivity factor along the current radial (as shown in Equations (1)-(2));

**dZ**, indicating the consistency of echo power in the adjacent range bins along the current radial (as shown in Equations (3)-(5));

**TDBZ** (unit: dB²), expressing the local textural consistency of reflectivity along the radial (as shown in Equation (6));

**SPIN,** representing the sign changes of adjacent reflectivity factors within a local area (as shown in Equations (7)-(8)).

$$R_{\rm REF} = \frac{\sum_{i=0}^{N_{\rm R}} N_{\rm Z}}{N_{\rm R}} \times 100\% \tag{1}$$

$$N_{\rm Z} = \begin{cases} 1 & Z_{i,j} = Val \\ 0 & Z_{i,j} \neq Val \end{cases} \tag{2}$$

$$B_{i,j} = Z_{i,j} - 20 \lg R_{i,j} \tag{3}$$

$$\overline{B} = \frac{\sum_{N_{R*0.9}}^{N_R} B_{i,j}}{N_R * 0.1} \tag{4}$$

$$dZ = B_{i,j} - \overline{B} \tag{5}$$

$$T_{\text{DBZ}} = \frac{\sum_{j=-5}^{j=5} (Z_{i,j} - Z_{i,j+1})^2}{11}$$
 (6)

$$M_{S_{\text{PIN}}} = \begin{cases} 1 & \frac{\left| Z_{i,j} - Z_{i,j-1} \right| + \left| Z_{i,j+1} - Z_{i,j} \right|}{2} > Z_{\text{thresh}} \\ 0 & \frac{\left| Z_{i,j} - Z_{i,j-1} \right| + \left| Z_{i,j+1} - Z_{i,j} \right|}{2} \le Z_{\text{thresh}} \end{cases}$$
(7)

$$S_{\text{PIN}} = \sum_{j=-5}^{j=5} M_{S_{\text{PIN}}} \tag{8}$$

In the equations, Zij (unit: dBZ) is the reflectivity factor at a certain range bin, Val is the effective detection value (unit: dBZ), Rij is the distance between the range bin and the radar (unit: km), NR is the number of range bins for the reflectivity factor, and  $Z_{thresh}Z_{thresh}$  is the

threshold for changes in the reflectivity factor between range bins.

For specific technical details, please refer to the following literature: Wen Hao, Zhang Lejian, Liang Haihe, Zhang Yang. 2020. "Radial interference echo identification algorithm based on fuzzy logic for weather radar." Acta Meteorologica Sinica, 78(1): 116-127. We will add this reference in the revised version of the manuscript.

**Changes in manuscript:** We have added a reference in line 341 of the revised manuscript. For a detailed description of the method, please refer to this reference.

• **P.14 Figure 12**: The left and right panels may be reversed—the left appears to be quality-controlled. Also, clarify which radar (and frequency band) was used to generate these results.

**Reply:** We will switch the order of the two panels in Figure 12 in the revised manuscript and add specific information about the radar type and frequency band.

**Changes in manuscript:** We have made revisions in the revised version; please refer to Figure 14, and the radar numbers are clarified in line 340.

P.14 Figure 13: Specify which radar was used for these results.

**Reply:** We will add a description of the radar parameters used in Figure 13 in the revised manuscript.

**Changes in manuscript:** We have made revisions in the revised version; please refer to Figure 14.

• **P.17 Figure 16**: The axis labels are too small to read. Also, it is unclear which radar the bias was calculated from.

**Reply:** When analyzing ground-based consistency, we set a distance threshold between adjacent radars (for example, 200 km between S-band radars), so any two radars within this threshold can be paired for matching. For Radar 1, if it can be paired with five surrounding radars, we calculate the bias between Radar 1 and each matched radar for every volume scan according to the method described in the paper. After one volume scan, Radar 1 will have five comparison results, and we take the mean of these five results as the final result for Radar 1 at that time. In this way, if Radar 1 has a significant systematic bias, it will be reflected in the bias result. If the standard deviation is large, it indicates that the observations from this radar are more dispersed and that further calibration and detailed analysis of the hardware are necessary.

**Changes in manuscript:** We have added supplementary explanations in lines 357–362 of the revised manuscript and have redrawn the figure. Please refer to Figure 17.

• **P.17 Figure 17**: The caption text within the figure is obscured by the data points.

**Reply:** We will revise Figure 17 in the updated manuscript to ensure that the caption text within the figure is clearly visible and easy to read.

**Changes in manuscript:** We have redrawn the figure and placed the deviations below each subplot. Please refer to Figure 18.

• P.20 L.362: The abbreviation "SC" is undefined and should be explained.

**Reply:** SC is a model of S-band radar used in operational applications; we will provide an explanation of this abbreviation in the revised manuscript.

**Changes in manuscript:** We have provided an explanation in line 438 of the revised version.

• **P.21 Figure 22**: This figure would be more informative if the x-axis used a time scale. **Reply:** The horizontal axis in Fig. 22 represents the number of volume scans. Since results are only calculated when weather events occur and meet the threshold, the horizontal axis does not correspond to continuous time. Therefore, we chose to use the sample size to represent the horizontal axis.

---

## Author Response (AR2)

**Response to review comments-1**

There are still a few minor issues that the authors should address in the final revision.

- 1. The full name of PMR should be given at its first occurrence in line 71 of the manuscript. **Reply:** It has been added at line 72 in the revised manuscript.
- 2. In line 86 of the manuscript, please check whether a hyphen should be used between "phase" and "array", and ensure consistency with the rest of the manuscript.

**Reply:** It has been added at line 86 in the revised manuscript.

- 3. In the legend of Figure 2, using "band" would be more appropriate than "type". **Reply:** It has been re-plotted and replaced in the revised manuscript.
- 4. In line 145, the manuscript mentions using data with elevation angles less than 4.5 degrees. What is the reason for this choice? Is it for computational efficiency, or are there other considerations?

**Reply:** In AWRCM, we only used data from the lowest five layers, mainly for computational efficiency. This is especially relevant during widespread stratiform precipitation events, where there are many overlapping points between adjacent radars, which would significantly reduce subsequent computational efficiency.

We have added a explanation of this point at line 94 in the revised manuscript.

5. The reference currently in line 254 should be moved to line 151, as this is the first time the threshold parameter is mentioned after its description has been added.

Reply: It has been adjusted in the revised manuscript.

**Response to review comments-2**

Thanks for the revision and answers to comments. They are appreciated.

The paper has improved, but you still miss to discuss obvious features in some of the plots, as detailed below. There are also som minor things I have noted:

**Reply:** Thank you for your further comments on the manuscript. We have carefully reviewed them and made the corresponding revisions. Our detailed responses are as follows.

1.I 267 please name the SNR thresholds

**Reply:** We have added the clarification at line 303 in the revised manuscript.

2.caption fig 10: please indicate what kind of correction has been applied.

Reply: We have added this clarification to the caption of Figure 10 in the revised manuscript.

3.I 314: briefely explain how the fuzzy logic classifier to identify interferences, since the reference is in Chinese and thus not accessible to at least some of the readers.

**Reply:** We extract four physical parameters that characterize radial interference echoes: DB, representing the consistency of echo power between adjacent range gates along the radial; RREF, representing the spatial extent of the reflectivity factor along the radial; TDBZ (units: dB²), representing the texture consistency of the local reflectivity factor along the radial; and SPIN, representing the sign change of adjacent reflectivity factors within a local region. Based on the probability distributions of these parameters, we construct corresponding membership functions and a binary (0–1) decision criterion for radial interference echoes. The criterion values are then combined via a weighted summation, and any point whose aggregated value exceeds a threshold is identified as a radial interference echo and removed.

We have added the clarification between I 315-322 in the revised manuscript.

4.caption fig 15: please explain the legends in the plots (eg. GB). Plots should be understood without going to the text to check for abreviations.. Make sure you do this for every figure in this paper.

**Reply:** We revised the legends of Figures 15, 16, 19, and 20 to use the names of the two methods for definition. In addition, we replaced the descriptions of the two analysis methods in Section 3.2 with their abbreviations. This makes the sources of the results more explicit and helps readers better understand the analysis.

5.I 353: how do you define an observation time? do you require a certain sample size? They are all characterized by the same meteorological conditions. If you have defined it somwhere, please repeat it here so that the reader can more easily follow your discussion.

**Reply:** "observation times" refers to a single complete volume scan, which typically takes about 5–6 minutes. The analysis period shown in Fig. 17 spans January to October 2024. After applying the screening criteria described earlier, the analysis primarily focuses on stable precipitation targets; however, different stations experienced different weather processes. In addition, when generating the bar charts for the above statistics, we only selected results where the sample size in the overlapping areas of neighboring radars exceeded 200 to ensure

the stability of the results. In the subsequent analysis of single-station, single-time cases, we did not impose this constraint in order to preserve as much detail as possible in the results. We have add these clarifications in line 353-362 in the revised version to help readers better understand the context.

6.Fig 18: how do you explain the 2 modes you can generally see in the plots? I haven't seen a discussion in the text, or did I missed that?

First mode seem to suggest that the bias of the radar increases with reflectivity. Does it suggest, that there is a problem with the attenuation correction? Such a discussion of an obvious feature should be included.

**Reply:** Figure 18 shows the satellite–ground analysis results for four S-band weather radar stations, where attenuation correction is not an issue. You may be referring to Figure 22. As you noted, the attenuation correction for the X-band phased-array radars is not very satisfactory; even after correction, the reflectivity remains biased low, and the bias increases with increasing reflectivity factor. We will add these points between I406-409 in the revised manuscript.

7.I 368 reference to fig 19: system 12 and 13 seem quite good in term of the bias for both methods. System 5 has a large bias. Especially for latter large bias you don't give an explanation. Is there a hardware problem, assuming the systems are identical (are they?)

**Reply:** The bias observed for System 5 is indeed pronounced. Our preliminary assessment points to a potential radar hardware issue. As this station is not part of the national operational assessment network, we are unable to implement centralized calibration and quality control procedures. Nevertheless, we have notified the provincial meteorological bureau to initiate a thorough investigation.

8.I 385 in reference to figure 21: you show individual observation time. Please give the definition of an observation time in the caption. Is it day of observation meeting certain conditions.

**Reply:** Individual observation time also refers to a single complete volume scan, typically lasting about 5–6 minutes. In our analysis, we did not specifically select observation days under particular conditions because both methods are engineered to run automatically. Each volume scan is processed, and target classification and filtering are implemented within the algorithms.

We have addressed this at line 391 in the revised manuscript.

9. Figure 22: radar in the right-hand figure behaves much different than the radar in the left-hand figure (which looks quite reasonable). A discussion is missing

**Reply:** Our analysis indicates that the satellite–ground comparison results for the radar on the left side of Fig. 22 are mainly concentrated in August 2024, whereas those for the radar on the right span June–September. The longer time window entails greater diversity in precipitation types, which may lead to differences in the satellite–ground scatter distributions. That said, this conclusion requires further verification with more observational data. We have

added a preliminary discussion in line 401-405 in the revised manuscript.

10. Figure 24: the figure after calibration: seems like the sample size and especially cases with reflectivities larger than 30 dB are missing; so I would say the suggested decrease in bias after calibration cannot be made unless you have comparable sample size and spread in meteorological situations.

**Reply:** Indeed, as you pointed out, the previous satellite—ground comparison after radar calibration was based on a relatively small sample size and lacked data above 30 dBZ. In the revised manuscript, we have extended the analysis period to July 2025, which increases the sample size compared with the original submission. We agree with your assessment: since this is a case-based analysis, it is still premature to conclude that "the bias decreases after calibration." We will revise the corresponding statement accordingly in lines 424–431 of the revised version.